# A Bio-Inspired Probabilistic Neural Network Model for Noise-Resistant Collision Perception

**DOI:** 10.3390/biomimetics9030136

**Published:** 2024-02-23

**Authors:** Jialan Hong, Xuelong Sun, Jigen Peng, Qinbing Fu

**Affiliations:** Machine Life and Intelligence Research Centre, School of Mathematics and Information Science, Guangzhou University, Guangzhou 510006, China; 2112115061@e.gzhu.edu.cn (J.H.); xsun@gzhu.edu.cn (X.S.)

**Keywords:** collision perception, bio-inspired, probabilistic neural network, noise resistance, LGMD

## Abstract

Bio-inspired models based on the lobula giant movement detector (LGMD) in the locust’s visual brain have received extensive attention and application for collision perception in various scenarios. These models offer advantages such as low power consumption and high computational efficiency in visual processing. However, current LGMD-based computational models, typically organized as four-layered neural networks, often encounter challenges related to noisy signals, particularly in complex dynamic environments. Biological studies have unveiled the intrinsic stochastic nature of synaptic transmission, which can aid neural computation in mitigating noise. In alignment with these biological findings, this paper introduces a probabilistic LGMD (Prob-LGMD) model that incorporates a probability into the synaptic connections between multiple layers, thereby capturing the uncertainty in signal transmission, interaction, and integration among neurons. Comparative testing of the proposed Prob-LGMD model and two conventional LGMD models was conducted using a range of visual stimuli, including indoor structured scenes and complex outdoor scenes, all subject to artificial noise. Additionally, the model’s performance was compared to standard engineering noise-filtering methods. The results clearly demonstrate that the proposed model outperforms all comparative methods, exhibiting a significant improvement in noise tolerance. This study showcases a straightforward yet effective approach to enhance collision perception in noisy environments.

## 1. Introduction

Collision detection and avoidance represent substantial challenges in the domain of autonomous navigation for robots, vehicles, and unmanned aerial vehicles (UAVs) [1]. Currently employed sensor approaches, such as radar [2], infrared [3], laser [4], and their combinations, have gained extensive usage in mobile machines. Nevertheless, these methods exhibit limitations concerning their reliability, system complexity, and energy efficiency. Furthermore, physical sensors, especially those pertaining to visual modalities, frequently face difficulties in complex dynamic environments, primarily due to the presence of noisy signals.

While computer vision techniques driven by deep learning have demonstrated a promising performance [5], these methods exhibit heavy reliance on large-scale datasets for training, demanding significant computational resources, and struggling to generalize to new scenarios. Additionally, they are often limited in their capability to handle real-time visual processing such as embedded vision systems. To address the need for data-free and energy-efficient motion perception methodologies, researchers have increasingly turned their attention to biological visual systems. They harness bio-inspired modeling and biomimetic approaches to construct dynamic vision systems [1,6,7,8], as well as advanced sensor strategies [9]. Notably, within the locust’s visual pathways, a pair of motion-sensitive detectors, LGMD1 and LGMD2, specialized in looming perception, have been identified, each exhibiting specific selectivity [10,11,12,13]. These biological and computational studies have showcased the considerable potential of bio-inspired visual systems for collision perception, offering energy efficiency and reliability for real-world applications. However, it is essential to highlight that due to their computational simplicity, LGMD-based models often face challenges in collision perception within noisy visual scenarios, leading to unexpectedly high false positives. In contrast, biological visual systems have evolved over hundreds of millions of years, rendering them highly robust in coping with noise.

In biological systems, neuronal spike generation is inherently stochastic [14]. This process is influenced not solely by signal transmission but also by factors such as ion channel states [15], protein synthesis [16], degradation, and specific physical connection characteristics [17]. Therefore, to align with biological principles, it is crucial to consider the uncertainty in neural signal transmission within neural network models. Incorporating probability into neural networks provides a suitable approach to capture this inherent feature of neural signal processing. In this context, several noteworthy studies have already demonstrated the advantages of integrating probability into third-generation neural networks, specifically spiking neural networks, enabling applications in classification, pattern recognition, and various other fields [18,19,20].

In contrast, existing biologically inspired motion perception neural networks are deterministic systems that produce identical responses to identical visual input stimuli, contradicting established biological theories. In biological systems, specifically, neuronal spike emissions are stochastic, leading to variable responses in organisms exposed to the same stimuli [21,22,23]. Building upon the innovative concepts of probabilistic spiking neural networks, we recognize that noise in motion signals inherently possesses randomness. In addressing the noise-related challenges in current modeling endeavors, we endeavor to integrate a probabilistic model into a classic LGMD neural network model and assess its effectiveness in mitigating noise.

In this study, we present the probabilistic LGMD (Prob-LGMD) model, which introduces probabilistic signal transmission, interaction, and integration into the classical LGMD-based multi-layered neural network. Our primary objective is to investigate its effectiveness in detecting colliding objects within various noisy scenarios, including complex ground-vehicle and air-vehicle scenes. These scenes are also augmented with artificial noise. The proposed probabilistic model preserves the foundational framework of the previous four-layered neural network model, encompassing photoreceptor cells, excitatory cells, inhibitory cells, summation cells, and an LGMD cell. However, it injects uncertainty into signal transmission between layers, interactions between excitatory and inhibitory cells to establish collision selectivity regarding the expansion of moving objects, and the integration of signals from the entire dendritic area into the LGMD output neuron. More specifically, we introduce random variables to determine the connection weights between layers. These variables independently follow a Bernoulli distribution with the same probabilistic parameter. Through systematic experiments, our proposed Prob-LGMD effectively and robustly extracts collision information from complex, noisy scenarios, consistently outperforming comparative methods.

Section 2 provides a brief review of related works. Section 3 presents the proposed model, including its formulation and parameter settings. Section 4 outlines the experimental setup and the metrics used for evaluation. Section 5 details the experiments conducted and analyzes the results. Section 6 offers the concluding remarks for this study. The abbreviations employed throughout this paper are summarized in Table 1.

## 2. Related Work

Within this section, we will discuss the most relevant research, including (1) probabilistic spiking neural models (pSNM); and (2) bio-inspired neural models for collision perception, which encompasses LPLC2, LGMD1, LGMD2, and MLG1 neural networks.

### 2.1. Probabilistic Spiking Neural Models

Spiking neural networks (SNNs) represent the third generation of artificial neural networks and hold great promise in addressing complex real-world challenges in both pattern recognition and motion perception [24,25,26]. A variety of spiking neuron models and SNNs have been developed, including the Hodgkin–Huxley models [27], spike response models [28], integrate-and-fire models [29], Izhikevich models [30], and others [31]. However, it is worth noting that the majority of SNN models are deterministic and have limited efficacy in addressing complex engineering problems, despite their utility in modeling biological neurons. Consequently, there is a need for novel SNN models that are capable of capturing the non-deterministic characteristics of brain signal processing. Such models are essential for effectively replacing existing ones in large-scale simulations [32].

Indeed, neurons are influenced not only by input signals but also by various factors such as gene and protein expression, physical properties, the synaptic probabilities of receiving spikes, neurotransmitters, and probabilities of ion channel openings, among others, at any given time [33]. Given the inherently stochastic nature of spike processes in biological neurons, it is fitting to seek new inspiration to enhance the current neural network models and explore the potential of probability-based SNNs for broader applications. In pursuit of this objective, Kasabov et al. introduced a groundbreaking probabilistic spiking neural model (pSNM) [34]. This model stores information in connection weights and probability parameters associated with spike generation and propagation. Building upon the pSNM framework, they developed an integrated SNN computational neurogenetics model by integrating gene interaction parameters with neuronal spiking activity, resulting in an integrated-pSNN [35]. This innovative approach has the potential for the computational modeling of cognitive functions and neurodegenerative diseases. These remarkable contributions served as inspiration for us to incorporate probabilistic models into the computational modeling of motion perception neural networks. By simply integrating the probabilistic model with a classic collision perception neural network [36], we observed its effectiveness in mitigating the impact of noise.

### 2.2. Bio-inspired Collision Perception Models

Over millions of years of evolution, many animals have developed a critical ability to evade rapidly approaching predators or threats by leveraging their efficient and robust visual systems [37,38,39]. These visual systems empower them to perceive looming objects with precision and timeliness. Among the visual motion detectors identified in invertebrates, the lobula giant movement detectors (LGMDs) [10,40], the lobula plate tangential cells (LPTCs) [41], and the small target motion detectors (STMDs) [42] have been the subject of extensive research and modeling in recent decades.

In the context of the locust’s visual system, two distinct LGMD neurons have been identified within the neural pathway responsible for processing natural visual cues. These neurons primarily respond to objects in motion that are moving in depth, signaling potential collisions. The LGMD1 neuron, initially identified as a motion detector nearly half a century ago, has been gradually recognized for its remarkable ability to exhibit the highest frequency of spike firing during collision events, while displaying only weak and brief activation in response to other categories of movements. Over time, numerous computational models have been developed to replicate the functionality of LGMD1 for collision perception [1,36], with a typical neural network model illustrated in Figure 1a. In proximity to LGMD1, the LGMD2 neuron shares many selective features for approaching objects but exhibits a distinctive preference for darker objects approaching the visual field. The numerical modeling of its unique selectivity was introduced by Fu et al., as reviewed in [6]. This modeling approach involved the incorporation of parallel ON/OFF channels into the classic, multi-layered LGMD neural network model, as depicted in Figure 1b. In this model, the ON channels generate inhibitions and temporally delayed excitations, while the OFF channels produce direct excitation and temporally delayed inhibition, as illustrated in Figure 1b. These channels converge and combine their outputs in a nonlinear fashion.

In the visual systems of the crab *Neohelice*, researchers have discovered mono-stratified lobula giant type 1 (MLG1) neurons that demonstrate sensitivity to looming stimuli [43,44]. These neurons possess finely tuned capabilities for encoding spatial location information, distinguishing them from LGMD neurons. The MLG1 neuronal ensemble not only enables the perception of the location of a looming stimulus but also plays a crucial role in continuously influencing the direction of the crab’s escape movements. Furthermore, in the optic lobe of *Drosophila*, lobula plate/lobula columnar type II (LPLC2) neurons were identified, exhibiting highly selective responses to expanding objects originating from the center of the visual field [45].

While bio-inspired neural networks implementing looming perception neural circuits have made substantial contributions to address the collision perception challenges across various engineering applications, most models tend to be susceptible to false detections or even failure in the presence of noise. Therefore, enhancing the performance of existing models against noise represents a valuable area for exploration. Drawing inspiration from the probabilistic approach of the pSNM, we endeavored to introduce uncertainty into the collision perception model, which is based on the classical LGMD neural network model [36], as depicted in Figure 1a. To the best of our knowledge, this is the first model to incorporate indeterminacy into motion perception neural models, aligning more closely with biological theories. Through comparative experiments, we discovered that the proposed model exhibits significant improvements in noisy and realistic scenarios. In contrast to approaches involving the addition of complex components or extra layers to neural networks, this research demonstrates a simplified method for enhancing collision perception against noise. This approach may serve as inspiration for further modeling studies in the field of motion perception.

## 3. Model Description

In this section, we present a comprehensive description of the proposed neural network model, referred to as Prob-LGMD. The visual neural network consists of four layers, which include the photoreceptor layer (P), the excitatory layer (E), the inhibitory layer (I), the summation layer (S), and an LGMD cell serving as the output layer. It is important to note that this model is designed to process visual signals derived from video sequences as its input.

Distinguishing itself from previous modeling approaches, our methodology introduces random variables into the connections between various layers: the P layer and the E layer, the P layer and the I layer, the E layer and the S layer, the I layer and the S layer, and the S layer and the LGMD cell. These random variables share a common probabilistic parameter, denoted as “prob”, as illustrated in Figure 2. Consequently, the model incorporates uncertainty in signal transmission, interaction, and integration.

### 3.1. Temporal Processing

The difference in luminance of images captured by the photoreceptor cells of the retina is depicted in Figure 2. We denote the output of the photoreceptor cells as P(x,y,t). Consequently, the output of a cell in the P layer can be defined as:(1)$$P\left(x,y,t\right)=\sum_{i=1}^{n_{p}}p_{i}P\left(x,y,t-i\right)+\left(L(x,y,t)-L(x,y,t-1)\right)$$
where P(x,y,t) represents the change in luminance for pixel (x, y) at frame *t*, and P(x,y,t−i) represents the change in luminance for the same pixel at frame t−i. The indices *x* and *y* refer to the positions within the matrix. L(x,y,t) and L(x,y,t−1) denote the luminance at the current frame *t* and the previous frame t−1, respectively. The parameter $n_{p}$ specifies the maximum number of frames for which the persistence of the luminance change is considered. The persistence coefficient $p_{i}$, belonging to the interval (0,1), is computed as $p_{i}={\left(1+e^{ui}\right)}^{-1}$, where u∈(−∞,+∞) and *i* indicates the number of frames preceding the current frame *t*.

### 3.2. E-I Layer with Probability of Signal Transmission

The output of cells in the P layer serves as input to the excitatory and inhibitory cells in the subsequent layer. In our proposed model, a random variable is introduced to determine whether the excitations from the P layer cells are transmitted to the corresponding cells in the second layer, known as the E layer. Specifically, the excitation E(x,y) of an E cell adopts the value of the corresponding P cell with a certain probability, or takes the value of 0. The output of an E layer cell can be defined as follows:(2)$$E\left(x,y,t\right)=P\left(x,y,t\right)X_{P,E}\left(x,y\right)$$
(3)$$X_{P,E}\left(x,y\right)∼Bernoulli\left(prob\right)$$

The excitation E(x,y,t), corresponding to pixel (x,y) at time *t*, is determined by the random variable $X_{P,E}\left(x,y\right)$, which follows a Bernoulli distribution with a probability of prob for taking the value 1 and a probability of 1−prob for taking the value 0. Here, prob∈(0,1] represents the probability that the signal from the P layer is transmitted to the corresponding pixel (x,y) in the E layer. It is important to note that the random variables described below share the same interpretation.

After one-frame delay, the inhibitions from cells in the P layer are transmitted to their neighboring retinal counterparts in the I layer with equal probability. The output of an I layer cell can be defined as
(4)$$I\left(x,y,t\right)=\sum_{i=-1}^{1}\sum_{j=-1}^{1}P\left(x+i,y+j,t-1\right)\omega^{I}\left(i,j\right)X_{P,I}\left(i,j\right)$$
where I(x,y,t) is the inhibition corresponding to pixel (x,y) at current time *t*; $\omega^{I}$ is the local inhibition weight given by
(5)$$\omega^{I}=\left[\begin{array}{ccc} 0.125 & 0.25 & 0.125\\ 0.25 & 0 & 0.25\\ 0.125 & 0.25 & 0.125 \end{array}\right]$$

### 3.3. S Layer with Probability of Signal Interaction

A crucial element of LGMD-based models involves spatial-temporal interaction, specifically the competition between excitation and inhibition, which shapes the specific selectivity to looming objects. The excitation of the E cell and the inhibition of the I cell are randomly transmitted to the S layer with a certain probability. The outputs from the E and I layers are summed at the S layer cell, which can be defined as follows:(6)$$S\left(x,y,t\right)=E\left(x,y,t\right)X_{E,S}\left(x,y\right)-I\left(x,y,t\right)X_{I,S}\left(x,y\right)W_{I}$$
where $W_{I}$ is the inhibition weighting.

To filter out isolated excitations caused by background details, we apply a transmission coefficient and a scaling factor to the output of the S layer. The transmission coefficient is determined based on the surrounding activations of the S layer cell and can be calculated using a convolution kernel. The computation can be expressed as follows:(7)$$C_{e}\left(x,y,t\right)=\sum_{i=-1}^{1}\sum_{j=-1}^{1}S\left(x+i,y+j,t\right)W_{e}\left(i,j\right)$$
where $W_{e}$ represents the influence of neighboring cells, and this operation can be simplified as a convolution mask:(8)$$W_{e}=\frac{1}{9}\left[\begin{array}{ccc} 1 & 1 & 1\\ 1 & 1 & 1\\ 1 & 1 & 1 \end{array}\right]$$

And, the scale ω is computed at each frame with the following formula:(9)$$\omega =max\left(\left|{\left[C_{e}\right]}_{t}\right|\right)C_{\omega }^{-1}+0.01$$
where ${\left[C_{e}\right]}_{t}$ represents the passing coefficient matrix, and $C_{\omega }$ is a constant.

Then, the signal in the S layer is calculated as the following multiplication:(10)$$\tilde{S}\left(x,y,t\right)=S\left(x,y,t\right)C_{e}\left(x,y,t\right)\omega^{-1}$$Finally, we filter out the decayed excitation.
(11)$$\tilde{S}\left(x,y,t\right)=\left\{\begin{array}{ccc} \tilde{S}\left(x,y,t\right)\;, & \; & if\;\tilde{S}\left(x,y,t\right)\geq T_{s}\\ 0\;, & \; & \mathrm{otherwise} \end{array}\right.$$
where $T_{s}$ is a positive threshold.

### 3.4. LGMD Cell with Probability of Signal Integration

At frame *t*, the membrane potential $K_{t}$ of LGMD cell is defined as
(12)$$K_{t}=\sum_{x=1}^{C}\sum_{y=1}^{R}\tilde{S}\left(x,y,t\right)X_{S,LGMD}\left(x,y\right)$$
where *C* and *R* are the spatial dimensions of the input stimuli.

The output of the network transforms the membrane potential with a typical sigmoid function as
(13)$$\kappa_{t}={\left(1+e^{-\frac{K_{t}}{n_{cell}}}\right)}^{-1}$$
where $n_{cell}$ is the total number of the cells in the P layer. It is important to note that the membrane potential $\kappa_{t}$ per se represents the output of our proposed network in this modeling study. We did not involve any spiking mechanism but compare the membrane potential of different related methods in our experiments.

### 3.5. Setting the Network Parameters

It is important to note that the random variables mentioned above, as per Equations (Equation 2), (Equation 4), (Equation 6) and (Equation 12), are all mutually independent and share the same probabilistic parameter prob. All the parameters of the Prob-LGMD model, including the newly added probabilistic parameter, are listed in Table 2, and were adapted from [36]. It is worth noting that the Prob-LGMD model currently operates without any learning methods and processes visual signals in a feed-forward manner.

## 4. Experimental Design and Performance Metrics

To evaluate the effectiveness of the proposed model in noisy conditions, we conducted comparative experiments involving objects approaching the visual field in various scenarios. The ground- and air-vehicle scenes themselves are inherently complex, characterized by natural optic flows in real physical scenes. Additionally, we introduced binary and Gaussian noise to the input visual stimuli to pose a challenge to the tested models. Our objective is to assess whether a practical collision perception visual system can effectively respond to approaching targets in noisy environments. It is important to highlight that the selection of the parameter prob will be explained in the following subsection. Furthermore, in consideration of the non-deterministic nature of the Prob-LGMD model, we performed 20 replicate experiments and calculated both the mean and variance of the model outcomes to demonstrate its robustness.

### 4.1. Evaluation Criteria

In essence, we anticipate that an effective collision perception visual system should exhibit its strongest response when a collision is imminent, while remaining relatively quiet during other moments. The selection of the probabilistic parameter directly influences the performance of Prob-LGMD models in collision perception across various scenarios. Therefore, determining the appropriate probabilistic parameter, essentially selecting an optimal one, becomes a critically important task. Our ultimate objective is to ensure that the proposed model performs admirably in both noise-introduced physical scenarios and real vehicle scenarios. To assess the model’s performance under different probabilistic parameters (prob in Table 2), we introduce an indicator called the “distinct ratio” (DR), which is mathematically defined as follows:(14)$$DR=\kappa_{max}-\frac{\sum_{i}^{T}\kappa \left(i\right)-\kappa_{max}}{T-1}$$
where $\kappa_{max}$ is the maximum value of the output membrane potential of the collision moment. κ(i) is the membrane potential at frame *i*. *T* is the total time length of the input image sequences.

We establish that the moment at which the maximum membrane potential value occurs represents the collision signal, with all other frames denoting noisy signals. As a result, a higher detection ratio (DR) value signifies improved model performance in distinguishing non-collision motion signals from collision signals. In essence, the DR serves as a pivotal criterion for selecting the optimal probabilistic parameter for the proposed Prob-LGMD model.

### 4.2. Setting the Experiments

The experiments can be broadly categorized into two types. Firstly, we conducted tests in indoor, structured scenarios where we introduced two forms of artificial noise into the input stimuli. These stimuli consisted of approaching black and white targets within a real physical scene. Figure 3a–c and Figure 4a–c display the intermediate stages of the looming process. These frames correspond to selected moments from the original video and have been altered with salt-and-pepper noise as well as Gaussian noise, respectively.

Secondly, in outdoor, unstructured scenarios, we present results related to ground/air-vehicle collision events. These intricate scenarios consistently pose challenges to LGMD-based computational models, including both the classical LGMD1 model [36] and the more recent LGMD2 model [13]. To compare the performance of the proposed Prob-LGMD model with these two typical models in predicting imminent collisions within noisy scenarios, we conducted experiments using a few ground-vehicle crash datasets. These datasets consist of partial frames extracted from the original video, the video with added salt-and-pepper noise, and the video with added Gaussian noise, respectively. Samples of collision courses are depicted in Figure 5a–c, in Figure 6a–c, in Figure 7a–c, and in Figure 8a–c. For the air-vehicle scenario, we employed a quadcopter to approach a black balloon, and the input stimuli were recorded by the onboard camera. After collecting the data, we introduced salt-and-pepper and Gaussian noise to raise the challenge, as exemplified in Figure 9a–c.

## 5. Results and Analysis

In this section, we present the experimental results along with their analysis. We conducted a comprehensive series of experiments to compare the performance of the proposed model with the classic LGMD1 model [36] and the state-of-the-art LGMD2 model [13] across various scenarios, ranging from straightforward to intricate visual scenes. The primary objective is to illustrate how the proposed probabilistic model effectively addresses the issue of false positives in collision perception caused by noisy signals. The main results can be categorized into four parts:1.Results in simple indoor scenarios;2.Results in complex outdoor scenarios;3.Results in the selection of probabilistic parameters;4.Results of further investigations.

### 5.1. Results under Testing of Structured Indoor Scenes

Figure 3 and Figure 4 depict the model responses of the proposed Prob-LGMD as well as the comparative LGMD1 and LGMD2 models when presented with the same visual input stimuli in indoor scenes. In the experiment where the black ball approaches without artificial noise, it is evident that all three models effectively perceive the proximity of the target. Specifically, their responses peak significantly just before the black ball fills the camera’s field of view, indicating no significant differences in their performance. However, when noise is introduced into the original stimuli, all three models become more sensitive before the actual collision occurs. In the case of stimuli with added salt-and-pepper noise, the LGMD1 model can detect collisions but shows an overall increase in response values. At higher noise densities, the LGMD1 model fails to detect the proximity of the target, as its activities remain at very high levels. The typical threshold mechanism struggles to differentiate collision detection. On the other hand, the LGMD2 model demonstrates better noise resistance and can still perceive collisions to some extent, although not as effectively as the Prob-LGMD model. When subjected to stimuli with added Gaussian noise, the LGMD1 model exhibits false detections, while the LGMD2 model is suppressed due to strong local, lateral inhibition caused by noise owing to its adaptive inhibition mechanism [13]. In conclusion, the Prob-LGMD model maintains its capability to detect collisions even in the presence of noise, making it a robust choice for collision perception in noisy environments.

In the experiment where the white ball approaches without artificial noise, the visual contrast is reversed. In this scenario, all three models perform similarly to the previous situation, showing no significant differences in their performance. However, when artificial noise is introduced, all three models become more sensitive. These results are consistent with the analysis mentioned earlier.

### 5.2. Results under Testing of Complex Vehicle Scenes

Figure 5, Figure 6, Figure 7 and Figure 8 illustrate the responses of the three model groups under different datasets, each also affected by the two aforementioned types of noise. These scenarios inherently possess more complexity and dynamism. The results clearly indicate that these models face substantial challenges, as their responses exhibit significant fluctuations. In the absence of artificial noise, both the proposed Prob-LGMD model and the LGMD2 model effectively perceive collisions, while the LGMD1 model proves to be more sensitive to motion stimuli. Without artificial noise, the LGMD1 model displays an overall excessive response, resulting in false detections. In contrast, the LGMD2 model demonstrates some resilience to noise when compared to the LGMD1 model and can reach its peak response before the collision frame in certain datasets. The Prob-LGMD model, relative to the two comparative models, reliably detects collisions and exhibits considerably lower responses for non-collision frames. Accordingly, the proposed model successfully leverages noise resistance for enhanced collision perception.

More specifically, with the addition of salt-and-pepper noise, both the LGMD1 and LGMD2 models fail to detect collisions, while the Prob-LGMD model continues to successfully identify collisions. However, when Gaussian noise is introduced, the LGMD1 model exhibits excessively high responses, whereas the LGMD2 model responds prematurely and experiences suppression due to the temporal dynamic inhibition mechanism triggered by the artificial noise, as explained in [13]. Furthermore, the gray shadow accompanying the responses of the proposed model in all the resulting figures illustrates the variance of the outputs over 20 repeated experiments. The results from the ground-vehicle scenario tests confirm the robustness of the proposed probabilistic model in combating noise.

We also conducted a comparison of model performance in air-vehicle scenarios. Figure 9 illustrates the superior performance of the Prob-LGMD model in the presence of noise. Specifically, in the absence of artificial noise, all three models effectively respond to collisions, detecting the gradually approaching balloon in the vicinity of the vehicle. However, upon the addition of artificial noise, only the Prob-LGMD model maintains stable collision perception performance. The LGMD1 model becomes overly sensitive, while the LGMD2 model loses its perception capability.

### 5.3. Selection of Probabilistic Parameters

In our previous preliminary work [46], we confirmed that the proposed “distinct ratio” indicator follows a convex curve concerning the model’s probabilistic parameter. The peak of this curve signifies the optimal effectiveness of the Prob-LGMD model. Consequently, the focus of this paper is to establish the range of probabilistic parameter values using a larger dataset. Figure 10a illustrates the variation of the DR across twelve datasets of noisy vehicle scenes, while Figure 10b displays the variations across ten datasets of noisy physical scenes. The blue line represents the mean DR for each dataset, while the gray shaded area reflects the DR’s fluctuation within each dataset. The graphs indicate that the Prob-LGMD model performs optimally when the probability parameter is approximately 0.5. In contrast, both the LGMD1 and LGMD2 models maintain a constant DR since they lack a probabilistic parameter. However, the LGMD2 model exhibits a higher DR than LGMD1, indicating its superior noise resistance.

Moreover, we quantified the experimental results of the Prob-LGMD model using various datasets, and the findings are summarized in Table 3. This table contains the DR-values of the proposed model across 51 visual stimuli, each tested with different probabilistic parameters. The initial five datasets present the model’s performance when analyzing the approach of a black ball in the original video, as well as in videos with added salt-and-pepper or Gaussian noise. The subsequent twelve datasets showcase the model’s results in real car scenarios subjected to various noise conditions. Each value in the table represents the mean and standard deviation of the model’s performance across different probability parameters in repeated experiments. The values are rounded to two decimal places, with the optimal values highlighted in reddish-colored font.

### 5.4. Further Investigations

#### 5.4.1. Comparison with Engineering Techniques

In engineering, one common approach for mitigating noise involves employing signal filters to remove noisy data points. To assess the performance of the proposed Prob-LGMD model in comparison to traditional engineering techniques, specifically Gaussian and median filters for noise reduction, we preprocess the input videos and then input them into the conventional LGMD1 model. We subsequently compare the results between the Prob-LGMD model and the LGMD1 model after pre-processing with these filters.

Figure 11, Figure 12, Figure 13, Figure 14, Figure 15 and Figure 16 depict the outcomes under noisy conditions, encompassing the direct input of stimuli to the LGMD1 model, the Prob-LGMD model, and the input to the LGMD1 model post preprocessing with Gaussian and median filters. It becomes evident that collision perception can be achieved by preprocessing the input to the LGMD1 model with suitable filters. However, even in cases where these engineering methods are effective, the proposed probabilistic model consistently outperforms them in complex, noisy scenarios featuring various types of artificial noise. Consequently, these results further validate the efficacy and robustness of the proposed approach for noise reduction and enhanced collision perception in intricate dynamic scenarios.

#### 5.4.2. Noise Injection between Network Layers

Additionally, our previous preliminary work has already showcased the Prob-LGMD model’s proficiency in effectively detecting collisions within simple scenes featuring artificially introduced noise [46]. This paper extends the evaluation of model performance to complex scenarios involving physical scenes and car collisions, comparing it with the LGMD2 model. Furthermore, we introduced artificial noise into the datasets prior to feeding them into various models.

To characterize the inherent noise within neural information transmission, we injected noise into the intermediate four-layer structure of both the classical LGMD1 model and the Prob-LGMD model, allowing us to assess how these models responded to noise added layer by layer. Figure 17 and Figure 18 illustrate that, following the introduction of noise at each layer, the LGMD1 model exhibited an overall increase in response, resulting in premature responses. Conversely, the Prob-LGMD model maintained its ability to discern collision courses and withstand noise introduced at intermediate layers. Furthermore, we conducted experiments to add noise separately during signal transmission, signal interaction, and signal integration, and the Prob-LGMD model consistently exhibited robust collision perception.

### 5.5. Further Discussion

Furthermore, we identified a limitation of the proposed method during our experiments. We observed that the model’s noise resistance capabilities are moderate when the figure–ground contrast is relatively low. This highlights the need for further enhancements in the proposed model, particularly in terms of addressing visual contrast. Additionally, it is essential for the probabilistic model to take into account the spatial and temporal distribution of probabilistic parameters, with these probabilities evolving over time.

The dynamic vision systems inspired by insect motion vision have found successful applications in micro-robot and vehicle vision, enabling real-world motion perception [1]. These models can be effectively integrated with the proposed probabilistic model to mitigate the impact of noisy signals. Moreover, autonomous navigation can benefit from neuroscience research into insect navigational strategies, particularly those known for their energy efficiency. Recently, a bio-inspired collision perception model was combined with an insect-inspired navigation model that emulates a vector-based homing behavior [47]. This alignment with the probabilistic model concept has significant potential to enhance the development of home vectors for foraging tasks with collision avoidance, especially in complex and dynamic environments.

## 6. Conclusions

This study has introduced a simple yet effective method to enhance collision perception’s robustness in the presence of noise. The main findings of our research can be summarized as follows:The comparative LGMD2 model demonstrates effective collision detection performance in physical and complex vehicle collision scenarios, even providing early collision warnings. However, in the presence of noise, both the LGMD2 and LGMD1 models struggle to recognize potential collisions, while the proposed Prob-LGMD model consistently exhibits a stable collision performance and resistance to noise.When compared to traditional engineering denoising methods such as Gaussian and median filtering, the Prob-LGMD model demonstrates superior noise resistance.By introducing noise into intermediate neural network layers to simulate the noise inherent in neural signal transmission and interaction, the proposed probabilistic model successfully overcomes this challenge and maintains robustness against noise.Based on biological insights, it is imperative to transition from deterministic motion perception models to probabilistic models. There is a wealth of modeling work in the field of dynamic vision systems that could benefit from the incorporation of similar stochastic processes.

These findings contribute to the advancement of collision perception systems, particularly in complex and noisy environments, and open up new possibilities for probabilistic modeling in the domain of dynamic vision systems.

## Figures and Tables

**Figure 1 biomimetics-09-00136-f001:**
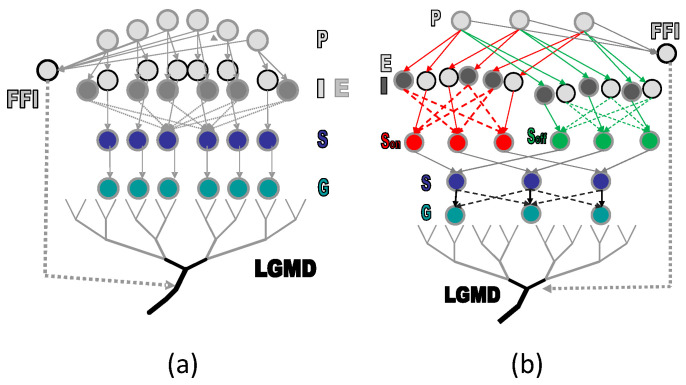
Schematic illustrations of typical LGMD neural network models are shown. The notations P, I, E, S, G, and FFI represent the photoreceptor, inhibition, excitation, summation, grouping layers, and the feed-forward inhibition pathway, respectively. The LGMD-based neural network models evolved from single-channel to dual-channel visual processing: (**a**) adapted from [36]; and (**b**) adapted from [13].

**Figure 2 biomimetics-09-00136-f002:**
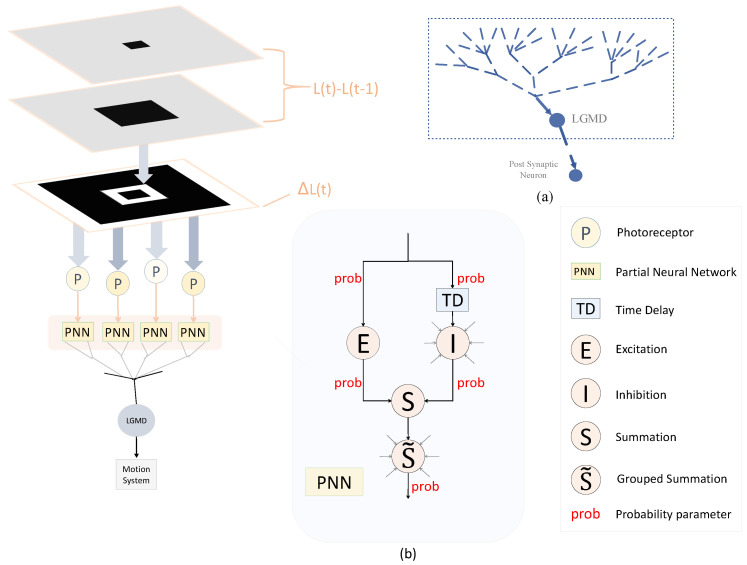
Illustrations of the proposed Prob-LGMD model: sub-figure (**a**) depicts the anatomical LGMD neuron and its dendritic structure; sub-figure (**b**) shows the basic network structure of Prob-LGMD. Note that we incorporate the probability parameters into the multi-layered neural network representing the non-deterministic nature of signal transmission, interaction, and integration. The model retrieves differential images from videos for visual processing. To account for the potential occurrence of negative values in the frame difference, we present the images using absolute gray-scale values.

**Figure 3 biomimetics-09-00136-f003:**
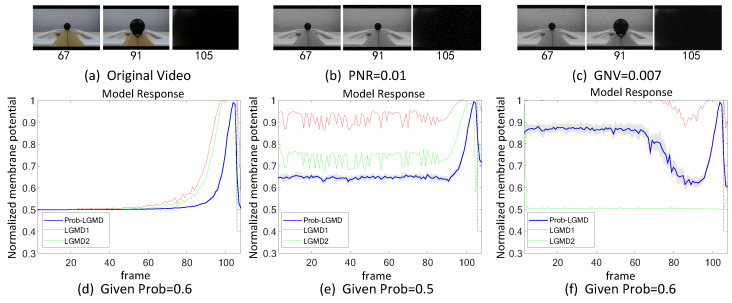
Sub-figures (**a**–**c**) show snapshots of the approaching black-ball in real physical scenes, as input stimuli with artificial salt-and-pepper noise, and Gaussian noise, respectively. ‘PNR’ and ‘GNV’ are abbreviations for pepper noise ratio and Gaussian noise variance, respectively. Sub-figures (**d**–**f**) represent the responses of the proposed model given different probability parameters, and comparative LGMD models against the input stimuli of (**a**–**c**), respectively. The gray-shaded area shows the variance of the Prob-LGMD model’s response across repeated trials, and the vertical dashed line indicates the ground-truth collision time in each scenario. The proposed method outperforms the comparative state-of-the-art LGMD models.

**Figure 4 biomimetics-09-00136-f004:**
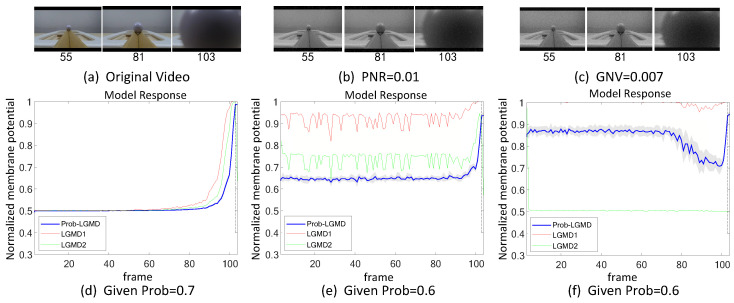
Sub-figures (**a**–**c**) show snapshots of the approaching white-ball in real physical scenes, as input stimuli with artificial salt-and-pepper noise, and Gaussian noise, respectively. Sub-figures (**d**–**f**) represent the responses of the proposed model given different probability parameters, and comparative LGMD models against the input stimuli of (**a**–**c**), respectively. Notations are consistent with Figure 3.

**Figure 5 biomimetics-09-00136-f005:**
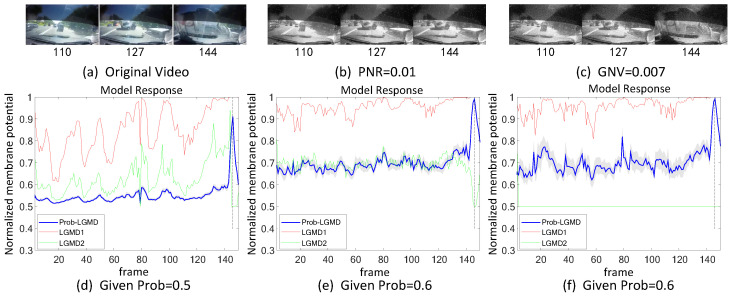
Results of the proposed method and two LGMD models against vehicle crash videos as input stimuli: this set of figures represents three datasets: the original data without noise, the data with salt-and-pepper noise, and the data with Gaussian noise. It can be observed that the leading vehicle collides with other vehicles from the left front and continues to approach the camera until a collision occurs. The Prob-LGMD model performs most robustly to perceive the crash: the membrane potential dramatically peaks only before the colliding moment.

**Figure 6 biomimetics-09-00136-f006:**
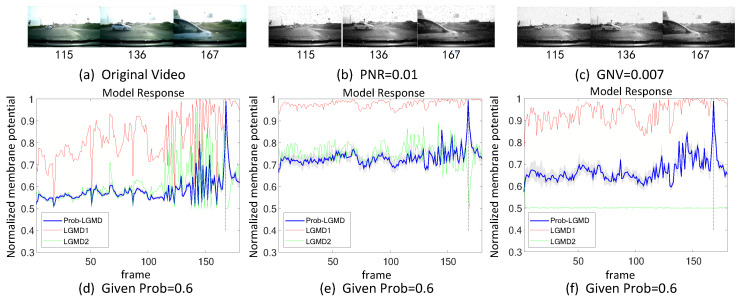
Results of the proposed method and two LGMD models against vehicle near-crossing videos as input stimuli: sub-figures (**a**–**c**) display snapshots of the original video, the video with salt-and-pepper noise, and the video with Gaussian noise, respectively. The Prob-LGMD model also performs well under different noisy circumstances.

**Figure 7 biomimetics-09-00136-f007:**
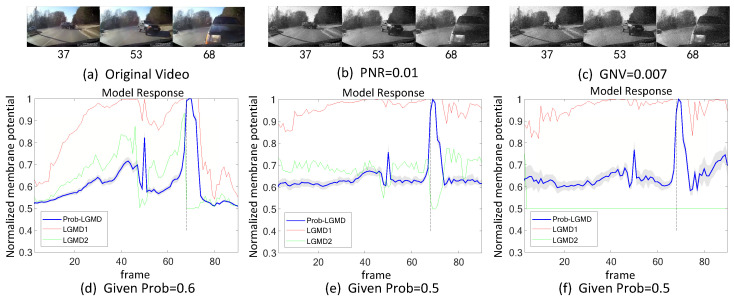
This set of visual stimuli illustrates the process of a car turning from the front side of the camera and colliding with it. Unlike previous stimuli, it gradually approaches from the right side, eventually filling the camera’s whole field of view.

**Figure 8 biomimetics-09-00136-f008:**
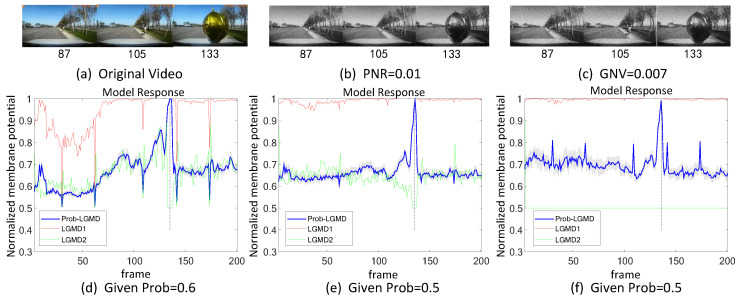
This set of visual stimuli shows a gradually approaching balloon directly to the frontal view of vehicle dashboard.

**Figure 9 biomimetics-09-00136-f009:**
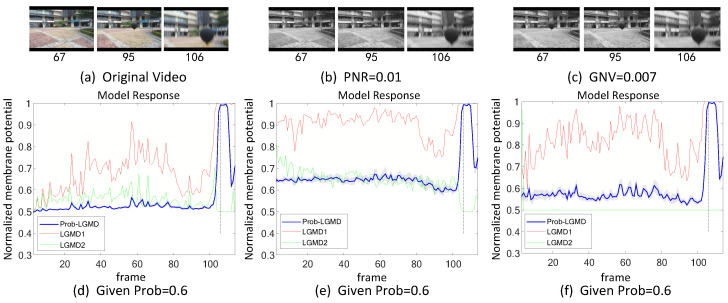
Sub-figures (**a**–**c**) show snapshots of a UAV approaching a black balloon, with artificial salt-and-pepper noise and Gaussian noise added as input stimuli, respectively. In this scene, the Prob-LGMD model also performs well.

**Figure 10 biomimetics-09-00136-f010:**
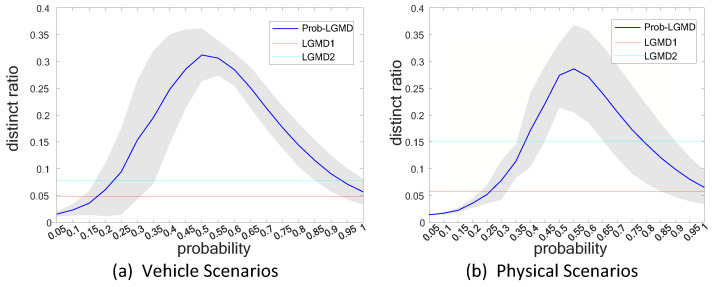
Statistical results of a distinct ratio calculated by Equation (Equation 14): sub-figures (**a**,**b**) display the DR of the proposed Prob-LGMD and two comparative models in vehicle-crash and ball-approaching physical scenes, respectively. The vehicle crash datasets include twelve video sequences. The physical datasets comprise ten approaching processes. The grey shadow represents the variance of the proposed model under different probability parameters, while the solid blue line represents the average DR. The experiments help us select an optimal probability parameter around the highest DR for optimizing the performance of our proposed model.

**Figure 11 biomimetics-09-00136-f011:**
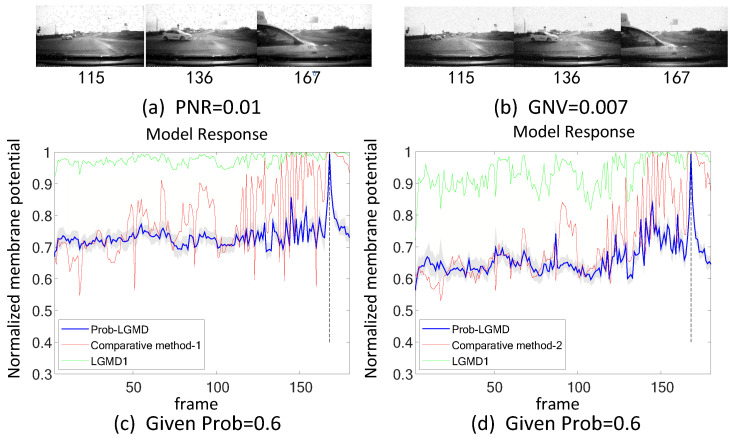
Sub-figures (**a**,**b**) display two sets of stimuli with different types of noise. Sub-figures (**c**,**d**) demonstrate the responses of the proposed Prob-LGMD, the comparative LGMD1, as well as the LGMD1 with a pre-filtering module under these two sets of stimuli. ‘Comparative method-1’ represents the approach of pre-filtering salt-and-pepper noise using median filtering, while ‘Comparative method-2’ represents the method of pre-filtering Gaussian noise using Gaussian filtering. These typical filters can improve the performance of the previous LGMD1 model to some extent whilst the proposed method still outperforms.

**Figure 12 biomimetics-09-00136-f012:**
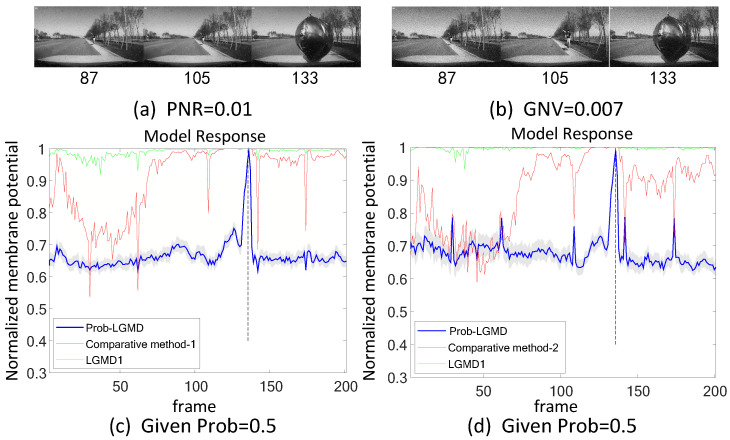
Responses of the proposed and the comparative methods under artificial salt-and-pepper and Gaussian noise in the colliding scenario: notations are consistent with Figure 11.

**Figure 13 biomimetics-09-00136-f013:**
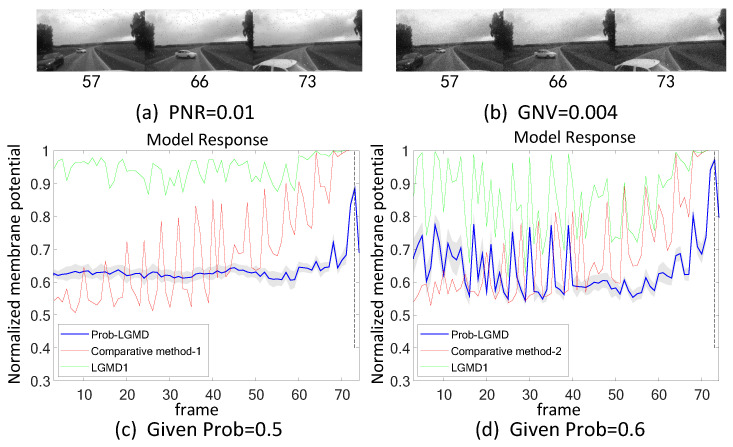
Responses of the proposed and the comparative methods under artificial salt-and-pepper and Gaussian noise in the colliding scenario.

**Figure 14 biomimetics-09-00136-f014:**
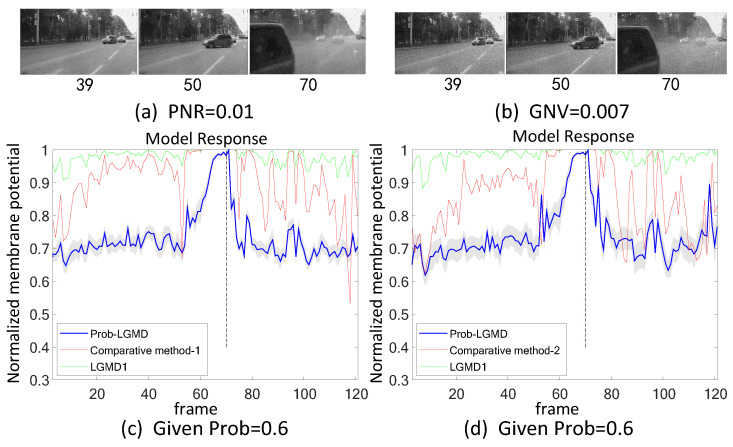
Sub-figures (**a**,**b**) show snapshots of the colliding scenario after adding noise, while sub-figures (**c**,**d**) correspond to the responses of the proposed model and comparative methods.

**Figure 15 biomimetics-09-00136-f015:**
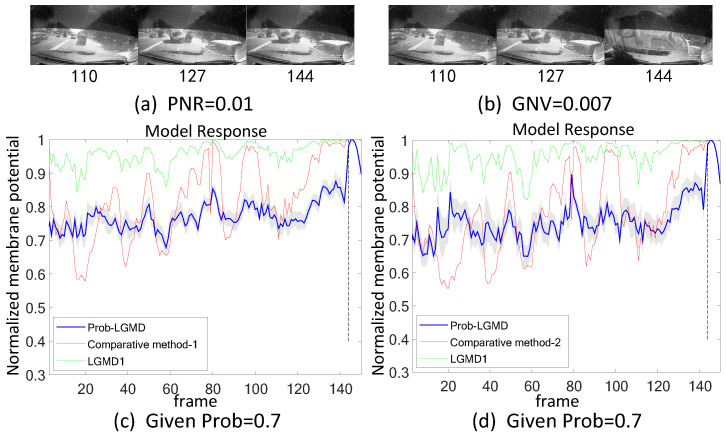
Sub-figures (**a**,**b**) show snapshots of the colliding scenario after adding noise, while sub-figures (**c**,**d**) correspond to the responses of the proposed model and comparative methods.

**Figure 16 biomimetics-09-00136-f016:**
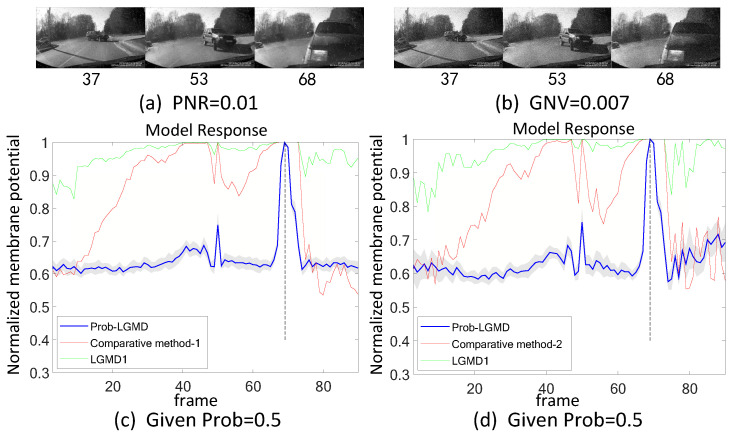
Sub-figures (**a**,**b**) show snapshots of the colliding scenario after adding noise, while sub-figures (**c**,**d**) correspond to the responses of the proposed model and comparative methods.

**Figure 17 biomimetics-09-00136-f017:**
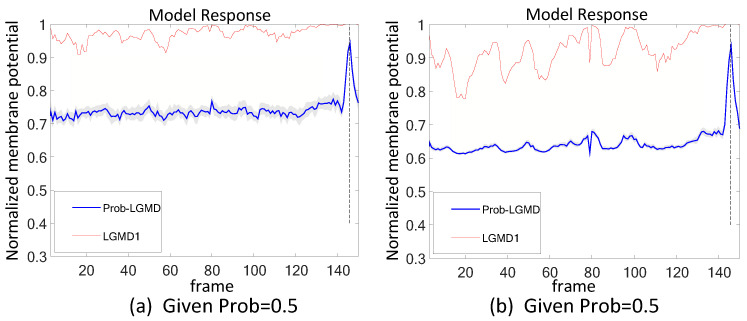
Investigation into introducing salt-and-pepper noise and Gaussian noise within the four-layered structure of the proposed neural network model: sub-figures (**a**,**b**) depict the responses of two models, the proposed Prob-LGMD and the comparative LGMD1, respectively. The input stimulus used accords with that in Figure 5a.

**Figure 18 biomimetics-09-00136-f018:**
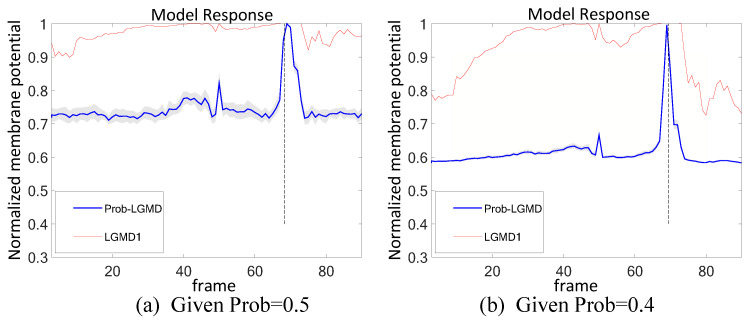
Investigation into introducing salt-and-pepper noise and Gaussian noise within the four-layered structure of the proposed neural network model: sub-figures (**a**,**b**) depict the responses of two models, the proposed Prob-LGMD and the comparative LGMD1, respectively. The input stimulus used accords with that in Figure 7a.

**Table 1 biomimetics-09-00136-t001:** Abbreviations in this paper.

Abbreviation	Full Name
LGMD	lobula giant movement detector
Prob-LGMD	probabilistic LGMD
pSNN	probabilistic spiking neural network
pSNM	probabilisitc spiking neural model
LPLC2	lobula plate/lobula columnar type II neuron
MLG1	monostratified lobula giant type I neuron
LPTC	lobula plate tangential cell
P,E,I,S	photoreceptor, excitation, inhibition, summation
DR	distinct ratio
PNR	pepper noise ratio
GNV	Gaussian noise variance

**Table 2 biomimetics-09-00136-t002:** Network parameters.

Parameter	Description	Value
$n_{p}$	Persistent luminance change duration	0∼2
$W_{I}$	Inhibition weight	0.3
$T_{s}$	Threshold in S-layer processing	30
$C_{\omega }$	coefficient in S-layer processing	4
prob	Probability parameter	0∼1
{C,R}	Spatial dimension of input stimuli	adaptable

**Table 3 biomimetics-09-00136-t003:** Quantification of distinct ratio of Prob-LGMD model under different datasets.

Dataset	Original			PNR = 0.01			GNV = 0.007		
	* **p** * ** = 0.5**	* **p** * ** = 0.6**	* **p** * ** = 0.7**	* **p** * ** = 0.5**	* **p** * ** = 0.6**	* **p** * ** = 0.7**	* **p** * ** = 0.5**	* **p** * ** = 0.6**	* **p** * ** = 0.7**
No.1	0.34 ± 0.01	0.44 ± 0.01	0.48 ± 0.00	0.28 ± 0.01	0.34 ± 0.01	0.31	0.15 ± 0.01	0.16 ± 0.01	0.17 ± 0.01
No.2	0.31 ± 0.02	0.42 ± 0.01	0.47 ± 0.00	0.16 ± 0.02	0.20 ± 0.01	0.18 ± 0.00	0.28 ± 0.01	0.33 ± 0.01	0.31 ± 0.00
No.3	0.32 ± 0.01	0.43 ± 0.02	0.48 ± 0.00	0.26 ± 0.02	0.32 ± 0.00	0.31 ± 0.00	0.15 ± 0.02	0.19 ± 0.01	0.18 ± 0.00
No.4	0.31 ± 0.02	0.43 ± 0.01	0.47 ± 0.00	0.13 ± 0.02	0.18 ± 0.01	0.16 ± 0.01	0.26 ± 0.01	0.32 ± 0.01	0.31 ± 0.00
No.5	0.31 ± 0.01	0.41 ± 0.01	0.47 ± 0.00	0.26 ± 0.01	0.33 ± 0.01	0.32 ± 0.00	0.13 ± 0.02	0.17 ± 0.01	0.16 ± 0.01
No.6	0.26 ± 0.03	0.36 ± 0.01	0.40 ± 0.00	0.19 ± 0.03	0.25 ± 0.01	0.23 ± 0.00	0.16 ± 0.02	0.20 ± 0.02	0.19 ± 0.01
No.7	0.46 ± 0.01	0.48 ± 0.00	0.46 ± 0.00	0.37 ± 0.00	0.34 ± 0.00	0.28 ± 0.00	0.33 ± 0.01	0.30 ± 0.00	0.24 ± 0.00
No.8	0.38 ± 0.02	0.44 ± 0.00	0.44 ± 0.00	0.31 ± 0.02	0.33 ± 0.00	0.30 ± 0.00	0.28 ± 0.02	0.30 ± 0.00	0.27 ± 0.00
No.9	0.24 ± 0.01	0.34 ± 0.02	0.38 ± 0.01	0.20 ± 0.02	0.24 ± 0.01	0.23 ± 0.01	0.17 ± 0.02	0.22 ± 0.02	0.23 ± 0.01
No.10	0.45 ± 0.01	0.44 ± 0.00	0.41 ± 0.00	0.38 ± 0.01	0.34 ± 0.00	0.29 ± 0.00	0.35 ± 0.01	0.31 ± 0.00	0.25 ± 0.00
No.11	0.38 ± 0.01	0.37 ± 0.00	0.30 ± 0.00	0.33 ± 0.01	0.29 ± 0.00	0.22 ± 0.00	0.35 ± 0.01	0.33 ± 0.00	0.26 ± 0.00
No.12	0.21 ± 0.02	0.28 ± 0.02	0.27 ± 0.00	0.29 ± 0.01	0.39 ± 0.01	0.41 ± 0.00	0.25 ± 0.02	0.29 ± 0.01	0.28 ± 0.00
No.13	0.39 ± 0.00	0.34 ± 0.00	0.28 ± 0.01	0.47 ± 0.00	0.44 ± 0.00	0.41 ± 0.00	0.40 ± 0.00	0.34 ± 0.00	0.28 ± 0.00
No.14	0.28 ± 0.01	0.26 ± 0.00	0.19 ± 0.00	0.37 ± 0.02	0.38 ± 0.00	0.33 ± 0.00	0.33 ± 0.01	0.30 ± 0.00	0.23 ± 0.00
No.15	0.16 ± 0.01	0.19 ± 0.01	0.21 ± 0.01	0.23 ± 0.02	0.35 ± 0.01	0.42 ± 0.01	0.18 ± 0.02	0.24 ± 0.02	0.26 ± 0.01
No.16	0.25 ± 0.02	0.27 ± 0.01	0.23 ± 0.00	0.32 ± 0.01	0.38 ± 0.01	0.37 ± 0.00	0.27 ± 0.02	0.30 ± 0.01	0.27 ± 0.00
No.17	0.24 ± 0.02	0.31 ± 0.01	0.31 ± 0.00	0.29 ± 0.02	0.38 ± 0.02	0.40 ± 0.00	0.24 ± 0.02	0.26 ± 0.01	0.24 ± 0.00

## Data Availability

The raw data supporting the conclusions of this article will be made available by the authors on request.

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
