# Peer review of "A Bio-Inspired Probabilistic Neural Network Model for Noise-Resistant Collision Perception"

_biomimetics, 2024, doi:10.3390/biomimetics9030136_

Round 1
Reviewer 1 Report
Comments and Suggestions for Authors
1. The motivation of the proposed method should be better justified.
It is argued in introduction that: "deep learning methods are heavily dependent of large-scale datasets for training, requiring substantial computational resources, hard to generalize to new circumstances, and also restricted to the capability of real-time visual processing." This seems not accurately to me. Tough deep learning requires large amount of training data. However, the widely using of deep learning in real-world applications proves the strong generalization ability of deep neural networks. And neural networks have been deployed to process image data and other data on various devices especially on mobile phones. This also indicates that the real-time visual processing is not an issue of deep learning anymore. So the motivation of the proposed method for data-free Prob-LGMD should be discussed in a more recent view.
2. The target problem is unclear. The goal of the noise-resistant collision perception is not given. More details such as the used signals and evaluation are missing.
3. There is no quantitative comparisons between the baseline methods and the proposed method. Only a few results on several images are given, which are not convincing. I would suggest evaluate the overall performance on a large data set.
Comments on the Quality of English LanguageThe English is overall good and easy to follow.
Author Response
Thanks very much for your valuable feedback on this research paper. Please refer to the uploaded file regarding our revision on this paper.

Reviewer 2 Report
Comments and Suggestions for Authors
1. Abstract needs improvement
2. Prob LGMD Model needs more clarity and explanation
3. Equation 4 needs more explanation and should have been supported by contents
4. Network parameters need more clarity and its relevance
5. Results of figure 6 need more explanation
6. improve result analysis
Comments on the Quality of English Languageneeds improvement
Author Response

(The authors gave the same response as above.)
